# Improving Aerosol Characterization Using an Optical Particle Counter Coupled with a Quartz Crystal Microbalance with an Integrated Microheater

**DOI:** 10.3390/s24082500

**Published:** 2024-04-13

**Authors:** Emiliano Zampetti, Maria Aurora Mancuso, Alessandro Capocecera, Paolo Papa, Antonella Macagnano

**Affiliations:** Institute of Atmospheric Pollution Research—National Research Council (IIA-CNR), Research Area of Rome 1, Strada Provinciale 35d, 9-00010 Montelibretti, Italyalessandro.capocecera@cnr.it (A.C.); antonella.macagnano@cnr.it (A.M.)

**Keywords:** aerosol, OPC, QCM, sensors, particulate matter

## Abstract

Aerosols, as well as suspended particulate matter, impact atmospheric pollution, the climate, and human health, directly or indirectly. Particle size, chemical composition, and other aerosol characteristics are determinant factors for atmospheric pollution dynamics and more. In the last decade, low-cost devices have been widely used in instrumentation to measure aerosols. However, they present some issues, such as the problem of discriminating whether the aerosol is composed of liquid particles or solid. This issue could lead to errors in the estimation of mass concentration in monitoring environments where there is fog. In this study, we investigate the use of an optical particle counter (OPC) coupled to a quartz crystal microbalance with an integrated microheater (H-QCM) to enhance measurement performances. The H-QCM was used not only to measure the collected mass on its surface but also, by using the integrated microheater, it was able to heat the collected mass by performing heating cycles. In particular, we tested the developed system with aerosolized saline solutions of sodium chloride (NaCl), with three decreasing concentrations of salt and three electronic cigarette solutions (e-liquid), with different concentrations of propylene glycol and glycerin mixtures. The results showed that the OPC coherently counted the salt dilution effects, and the H-QCM output confirmed the presence of liquid and solid particles in the aerosols. In the case of e-liquid aerosols, the OPC counted the particles, and the HQCM output highlighted that in the aerosol, there were no solid particles but a liquid phase only. These findings contribute to the refinement of aerosol measurement methodologies by low-cost sensors, fostering a more comprehensive understanding.

## 1. Introduction

An aerosol is a colloidal suspension of fine solid or liquid particles in a gas and has been investigated in several research fields. In medicine, it is mainly studied as a method for drug delivery [1]. In material and chemical science, it is studied as a method to produce or functionalize materials [2]. Aerosols, as well as particulate matter or suspended particulate matter (PMx), impact atmospheric pollution, the climate, and human health, either directly or indirectly. For these reasons, the number of publications on atmospheric aerosols has dramatically increased in the last decade [3]. Aerosols play an important role in environmental pollution, and the measurements of the particles’ sizes or their chemical composition are crucial for understanding atmospheric pollution dynamics [4,5,6]. Knowledge of aerosols is continually deepening thanks to scientific research, but concepts such as deliquescence transition, efflorescence, coagulation, nucleation, or hygroscopic growth of salt are not fully understood when the size is submicrometric [7].

In atmospheric pollution monitoring (or PM), aerosols are among the most important pollutants, and several devices are available to perform automatic measurements, such as scanning mobility particle sizer (SMPS), quartz crystal microbalance cascade, beta gauge, and other vibrating devices [8,9,10]. While these devices provide accurate and reproducible responses, they often occupy significant volumes, and their management costs are not sustainable for use in large-scale mapping applications. In fact, most of these mentioned measurement systems require a sampling system upstream of the instrument to treat the air sample (e.g., dryers, thermal conditioning sampler, filters, etc.), which cannot be easily miniaturized and have a non-negligible impact on costs and energy consumption [11,12]. For these reasons, the research community is addressing the study and development of low-cost sensors or sensor systems [13]. These devices, with their small size, low cost, and low power consumption, are suitable for application scenarios where a large number of sampling points are needed [14,15,16]. Optical particle counters (OPCs) and quartz crystal microbalances (QCMs) are low-cost devices widely employed to measure and study aerosols. OPCs count particles with an equivalent diameter in certain ranges using the laser scattering method [17]. QCMs measure the thickness or mass of collected substance on its electrode surface by ultrasonic vibration of the piezoelectric crystal [18,19,20]. These devices are widely used in biosensing, medical, space, and pollutant monitoring applications [21,22,23,24].

As described by Görner et al., a well-calibrated OPC is suitable for aerosol mass concentration monitoring in the workplace [25]. Additionally, in a paper by Hand, Jenny L., and Sonia M. Kreidenweis, OPC data were used to implement a new data analysis method to retrieve the refractive index and effective density from aerosol size distribution data [26].

However, measurements carried out only by OPC can be influenced by the different density or composition of the aerosol, particularly when data about mass concentration are desirable. In fact, to calculate the mass concentration of aerosols, the OPC uses a fixed particle mass density value that can vary depending on the aerosol composition [27,28]. QCM and OPC are powerful tools that can provide valuable information about the properties and behaviour of particles in a wide range of applications. Using them together can provide even more comprehensive insights. As described in the research paper by Kyeong-Rak Lee et al. [29], an OPC and QCM can be used together in certain applications to provide complementary information about the properties and behaviour of particles.

In this work, we propose combining a low-cost OPC with a modified QCM and integrating a microheater and a microresistance temperature detector on its surface (H-QCM) as reported in our previous paper [30]. The developed sensor system produces both particle counting and the possibility of obtaining information regarding the presence of liquid or solid phases in the analyzed aerosol sample. In particular, we tested the developed system with aerosolized saline solutions of sodium chloride (NaCl), with three decreasing concentrations of salt and three electronic cigarette solutions (e-liquid), with different concentrations of propylene glycol and glycerin mixtures. The results highlighted the possibility of using the QCM combined with OPC to determine the presence or absence of a solid phase in the tested aerosol.

This improvement could be applied in real cases where the measurement of aerosols (or PM) may not be entirely reliable using only low-cost OPCs, without any kind of sample processing system, due to the aerosol composition or interference agents such as fog or oil vapours.

## 2. Methods and Materials

### 2.1. Solid and Liquid Aerosols

An aerosol is a colloidal suspension of fine solid or liquid particles in a gas [6,31]. An example of a solid aerosol is aerosol salt. Specifically, NaCl aerosol refers to a type of aerosol containing sodium chloride particles generated via various methods such as salt sprays, nebulizers, or humidifiers. These particles vary in size, ranging from fractions of a sub-micrometre to a few millimetres, and they may be uniformly or unevenly distributed throughout the gas.

The salty particles of NaCl easily absorb moisture from the air, making them susceptible to deliquescence and efflorescence. These phenomena occur when the salty particles, now in a wet form, either dissolve into a liquid solution (deliquescence) or leave behind salt crystals on a solid surface as the water evaporates (efflorescence). The incidence of these dynamics depends on factors like humidity and temperature [32].

Instead, an example of a liquid aerosol is the aerosolization of a mixture of propylene glycol (PG) and vegetable glycerin (VG), commonly used as e-liquid for electronic cigarettes. This mixture (when vaporized) produces an aerosol that users inhale. Properties of this aerosol depend on the ratio of PG to VG in the e-liquid. PG produces small liquid particles and less dense aerosol than VG, while VG produces bigger particles and a denser aerosol [33,34].

The OPC is able to detect particles using light scattering and faces a fundamental limitation in differentiating the physical state of the particles. While it provides valuable information about particle size in aerosols, it cannot distinguish between solid and liquid particles solely based on light scattering [25,29].

To address this limitation, especially for the study of liquid and saline aerosols, a combination of the OPC and the H-QCM [30,35] could be very interesting.

The salt particles may initially be found in an aerosol phase, and as soon as they come into contact with the QCM surface, they may undergo phase transition and form a solid deposit.

According to Sauerbrey’s Equation (1), the deposition of salt particles on the QCM surface can lead to an increase in the mass of the QCM, which can be measured as a change in the resonant frequency of the crystal as follows:(1)$$\Delta f=-\frac{2\;fo^{2}}{A\;\sqrt{v_{q}\;\rho_{q}}\;}\Delta m=-\frac{C_{f}}{A}\Delta m$$

Δ*f* refers to the frequency shift due to the changing mass Δ*m*; *C_f_* represents a constant, where *f*_0_ is the fundamental resonating frequency; *v_q_* is the velocity of propagation of the transverse wave in the plane of the quartz; *ρ_q_* is the density of the quartz; and *A* is the effective area of the electrode [36].

The deposition of aerosol salt on an H-QCM can be influenced by various factors, including the concentration and size distribution of the salt particles in the aerosol, the relative humidity of the air, and the temperature of the H-QCM surface [35]. In fact, considering these parameters, phenomena like efflorescence and deliquescence could potentially occur.

Propylene glycol (PG) and vegetable glycerin (VG) can be mixed to produce a liquid aerosol that is detectable on a quartz crystal microbalance (QCM). In this case, detecting liquid particulate matter introduces additional considerations. Liquids may exhibit behaviours such as wetting or adherence, affecting how they distribute and adhere to the QCM surface. This can result in a more nuanced change in mass compared to solid particles. Additionally, it is important to note that under certain conditions, the liquid aerosol could evaporate from the QCM surface.

While the core principles of mass detection can be applied to both solid and liquid particulate matter on a QCM [37], their specific dynamics and interactions can differ. Understanding these subtleties is crucial for interpreting QCM data accurately in studies involving both solid and liquid aerosols to obtain analytical results.

### 2.2. Reagents and Samples Preparation

The chemicals used to prepare the samples, which included two different sets of solutions, were vegetable glycerol (VG, CAS No. 56-81-5), propylene glycol (PG, CAS No. 57-55-6), and sodium chloride (NaCl, CAS No. 7647-14-5) purchased from Merck (Darmstadt, Germany). A stock solution of NaCl (0.15 M, approximately 0.90%) at a physiological concentration (NaClphy) was prepared by weighing 2.2104 g of NaCl in 250 mL of distilled water (H_2_Odist) as the solvent. Two additional solutions were prepared by diluting the stock solution by half (NaCl 1:2, 0.075 M) and tenfold (NaCl 1:10, 0.015 M), respectively. After aerosolization, this set of solutions (NaClphy, NaCl1:2, NaCl1:10) simulated saline aerosols, where fine particles of NaCl were suspended in the air at different concentrations.

In addition, another set of three solutions was prepared using different ratios of propylene glycol (PG) and glycerol (VG): liq80:20, liq50:50, and liq20:80 of PG and VG, respectively. This time, liquid aerosols were generated, where fine particles of PG:VG were suspended in the air. To summarize, six samples were prepared, three of which are related to solid aerosol (NaClphy, NaCl1:2, and NaCl1:10) and three of which are related to liquid aerosol (liq80:20, liq50:50, and liq20:80). In order to observe the differences between liquid droplets and solid particulate in the samples, a solution of NaCl and one of e-liquid were nebulized over an optically polished quartz slice. Details of the nebulization system will be described in the following paragraph. Using an optical microscope (Leica DM2700 M, By Leica, Milan, Italy) with a 100×/0.85 magnification objective and a 22 mm field of view, liquid droplets of PG:VG were observed (Figure 1a), while solid particles of NaCl were observed (Figure 1b) after aerosolization on a quartz crystal slice. In particular, the photo on the right was captured after a heating treatment of the nebulized sample quartz. In fact, without heating, even the sample with NaCl appeared in the form of liquid droplets.

### 2.3. Measurement Setup

For these experiments, an OPC (OPC-N2 by Alphasense Ltd., Great Notley, Braintree Essex CM77 7AA, Rayne, UK) and an H-QCM were used together to provide complementary information about the properties of aerosols (e.g., liquid or solid phases). Figure 2 reports the scheme of the measurement setup used to perform all tests presented (Appendix A shows a photograph of the setup).

Samples were aerosolized using a piezo vibrating mesh nebulizer (1100 holes, 6 um of diameter, and 110 kHz of vibrating frequency) [38] placed inside a chamber (generation chamber) and controlled by a nebulizer control board. The OPC withdrew an aerosol sample (for a duration of 1 min) from the generation chamber using its integrated fan. Following this, the sampled aerosol reached the H-QCM connected to the output of the OPC by a suitable mechanical adapter (H-QCM chamber). The OPC data and the resonant frequency of H-QCM were acquired by the acquisition and control board. Finally, a PC unit was used to store all the data. The frequency shift Δ*F* = *f*(*t*) − *f*0 was used to signify output data of H-QCM where *f*0 was the resonant frequency obtained without any mass deposition. Appendix A shows a detailed view of the developed sensor system.

As previously described, the H-QCM, based on AT-cut quartz crystal, was used to measure the mass of the aerosol sampled by OPC. At the same time, the integrated heater of the H-QCM heated the collected mass on the surface of the sensor in order to discern the aerosol characteristics. The microheater was a double omega-shaped thin film (one on the top and one on the bottom of the crystal) that was connected to a temperature controller that measured the temperature and regulated the power supplied. The temperature of the heater was calculated by the controller using a calibration curve concerning the heater electrical resistance and temperature (measured by several micro-thermocouples during the calibration activities) [39,40]. During the heating, the crystal frequency changed following the behaviour of AT-cut crystal, and then, after the heating, the frequency returned to the previous value [41]. The graph reported in Appendix A illustrates the frequency shift observed at various temperatures provided by the integrated heater without deposited mass. The steps of temperature were set by the acquisition and control board to perform the calibration. A maximum value of $\bar{\Delta f}$ = 2909 Hz was reached for $\bar{T}$= 180 °C when 1.1 W of power was provided to the heater in the presence of an air flux produced by the OPC fan (dynamic behaviour of the frequency during the heating step is reported in the Appendix A).

## 3. Results and Discussions

The following results were obtained during the test following the nebulization of saline aerosol (solid aerosol), and PG/VG aerosol (liquid aerosol) will be presented. In particular, in Section 3.1 and Section 3.3, we reported the data obtained from the OPC, and in Section 3.2 and Section 3.4, the results concerning the H-QCM.

### 3.1. Measurements of the Saline Aerosol with OPC

When an aerosol containing salt particles is introduced to an OPC, the salt particles may be detected and counted as individual particles. The detection of salt particles depends on several factors including the size and concentration of the particles and the refractive index. Salt particles typically have a high refractive index compared to other aerosol particles, which can make them more easily detectable by an OPC [42,43].

The Count Mean Diameter (CMD) was derived from measurements taken with the OPC for the three solutions; it can be obtained by calculating a weighted average based on the number of particles in each size range. Then, CMD provides an estimation of the average size of saline aerosol particles, allowing us to gain a better understanding of the particle size distribution [44].

The counts related to the three NaCl solutions were plotted (Figure 3). Specifically, on the y-axis, normalized counts are plotted, and on the x-axis, the bins corresponding to different diameters are represented (Appendix A reports an example of the counts as a function of the aerodynamic diameter (d) of NaCl 1:10 and distilled water, H_2_Odist).

The blue histogram represents the size distribution of particles in the NaCl physiological solution where the calculated CMD is 2.90 ± 0.45 µm for this sample. The red histogram refers to the dimensional distribution of the NaCl_1:2_ solution, which has half the concentration compared to the physiological one. In this case, a CMD of 2.83 ± 0.44 µm was obtained. The graph of the latest solution diluted with a 1:10 ratio to the physiological one (NaCl_1:10_) is represented by the green histogram. The calculated CMD is 1.75 ± 0.39 µm, and this value slightly deviates from the first two, considering experimental error as well. This result may be related to the dilution of this latest solution compared to the initial one.

Both NaCl_phy_ and NaCl_1:2_ exhibit a CMD value that is comparable within experimental error, indicating a comparable size distribution. The particles in the more concentrated solution may initially have larger sizes due to the higher solute concentration. Consequently, during nebulization, the OPC may detect both larger and smaller particles, resulting in a broader size distribution. Conversely, in the case of the diluted solution, the particles may be initially smaller. As the droplets fragment during nebulization, predominantly smaller particles may be generated. The OPC is likely to primarily detect smaller particles, leading to a size distribution that is more concentrated around smaller diameters.

### 3.2. Measurements of the Saline Aerosol with H-QCM

The H-QCM offers a complementary approach, measuring frequency variations during the collection of aerosols to discriminate between solid and liquid phases. The collection of saline aerosol particles on its electrode induces a frequency shift correlated to the added mass, as shown in Equation (1). The H-QCM provides a real-time and sensitive method for monitoring the dynamics of saline aerosol particles. Furthermore, the heating process facilitated by the integrated heater enables the evaporation of the solvent (H_2_Odist) from the prepared solutions. This allows for the correlation of the frequency shift exclusively to the solid mass deposited on the quartz. This approach allows for the discrimination between the presence of solid and liquid particulates on the surface.

Figure 4 shows the frequency shift chronogram obtained for the NaCl_phy_ sample. Before turning on the heater, the observed Δf relates to an aerosol deposition, a mix of solid and liquid phases. In this state, the signal does not settle, unlike after heating. This could be due to the dynamic processes on the H-QCM surface, where both solid NaCl and liquid H_2_O are present. In fact, when salt arrives in the form of an aerosol, it will not be in its solid crystalline state but will be surrounded by a certain concentration of water [45]. This condition can result in surface dynamics that delay the stabilization of the signal. For instance, phenomena such as deliquescence and aggregation, where salt particles may cluster together or adsorb water on the surface, can occur. This can result in the formation of more complex structures or the creation of a thicker and more homogeneous layer. Furthermore, variations in surface tension may still occur, affecting the water’s ability to wet the surface or influencing other behaviours related to surface properties [46].

After three minutes, the surface heater was turned on, resulting in a rapid positive variation in Δ*f* and was turned off after two minutes (rapid decreasing and stabilizing at a frequency of Δ*f* = 354 Hz). The reached temperature resulted in the evaporation of water, and NaCl was present as a solid crystalline form. During this process, the compound may undergo changes in its crystal structure. The phenomenon of efflorescence is often associated with the evaporation of water containing dissolved salts, which leads to the crystallization of salts on the surface of the material. The frequency shift remains stable after the heating cycle. In fact, when heated, the bound water is released in the form of vapour, leaving behind anhydrous sodium chloride, which is devoid of water. A second measurement was performed consecutively, and once again, it was possible to observe the first and second rapid variation in the frequency following the activation and deactivation of the heater, stabilizing at a frequency of Δ*f* = 301 Hz. 

It is possible to note that before heating cycles in both measurements, the frequency shift is smaller compared to that observed after heating. This could be correlated with the sizes of the salt crystals that initially contain hydration water. The dimensions of some crystals might be larger than 2 microns and therefore may not be detected by the 10 MHz QCM, as discussed in our previous article [47]. Conversely, after heating, a larger frequency shift suggests the evaporation of H_2_O from the salt particle, resulting in smaller dimensions that can be detected by the QCM. This is consistent with the results obtained from the OPC, where the CMD for NaClphy was measured to be 2.90 ± 0.45 µm.

Appendix A present examples of chronograms of the NaCl_1:2_ and NaCl_1:10_ saline solutions. The behaviour of the NaCl_1:2_ solution is entirely analogous to that of the physiological solution, including the dimensional effect, consistent with the values obtained from the calculation of the CMD (2.90 µm and 2.83 µm, respectively). For NaCl_1:10_ aerosols, the majority of the crystals are presumably below 2 µm, as detected by measurements with the OPC, resulting in a CMD of 1.75 µm. For this reason, the decrease in Δf is related to the evaporation of water and, consequently, to the loss of mass from the quartz crystal surface (Equation (1)). The integration of the OPC and H-QCM enhances our understanding of the particulate matter an aerosol is composed of. The OPC, which relies on light scattering, may encounter challenges when dealing with salt particles, as discussed in the preceding paragraphs regarding humidity and aggregation phenomena. Meanwhile, the H-QCM, with its heated surface, provides real-time sensitivity and the ability to discriminate between solid and liquid phases. The analysis of NaCl solutions emphasizes OPC’s capability to detect diverse particle sizes, complemented by the H-QCM’s observations of frequency changes during nebulization and particle dynamics.

### 3.3. Measurements of the Liquid Aerosol with OPC

When a liquid aerosol passes through the OPC, the droplets interact with the laser beam, operating similarly to when dealing with saline aerosols.

This implies that measurements obtained using an OPC do not discriminate between aerosol phases. To obtain data that closely represents real conditions, a correction for the density of the different particulate matter detected is necessary. Indeed, one of the main limitations of the OPC is that particles of different substances may have different densities. This means that even if two particles have the same optical diameter, they could have different volumes or aerodynamic masses due to their density [48]. Also, in this case, the integration of an OPC with a H-QCM could enhance the understanding of the aerosol’s characteristics. 

In Figure 5, the green histogram shows the size distribution of the PG/VG liquid aerosol in an 80:20 ratio, with a found CMD of 0.90 ± 0.17 µm. The red histogram represents the dimensional distribution of the 50:50 PG/VG, resulting in a CMD of 1.01 ± 0.15 µm. In the last case, the blue histogram illustrates the normalized count distribution of the PG/VG 20:80, yielding a CMD of 1.13 ± 0.15 µm. The dimensional distribution is nearly similar for all three samples with different ratios of PG/VG, considering the obtained CMD value and the calculated errors.

### 3.4. Measurements of the Liquid Aerosol with QCM

The liquid aerosol induces a frequency shift of H-QCM related to the deposition of mass, although the liquid aerosol is expected to undergo natural evaporation (at room temperature) from the crystal surface over a specific time frame. However, by utilizing the integrated heater, it becomes possible to accelerate the evaporation of liquid with high evaporation temperature. This allows us to discern whether the phase of an aerosol detected by the H-QCM is solid or liquid, a distinction that is not achievable with a standalone OPC (Section 3.3).

In Figure 6, we present the resulting chronogram of two distinct depositions before and after heating. The aerosol generated within the nebulization chamber was aspirated for one minute by the pump of the OPC. Subsequently, the aerosol flow was directed onto the surface of the QCM. After three minutes of stabilizing the frequency shift, the heater was activated, reaching a temperature of about 170 °C in two minutes. This was followed by a waiting period of five minutes to ensure signal stabilization. The experiment reveals that before activating the heater, a frequency shift occurs that might be mistakenly associated with the presence of solid particulate (Δ*f* ≠ 0). However, during the heating process, the liquid aerosol droplets evaporate from the surface, restoring the initial resonance frequency. We obtained the same behaviour for the liq80:20 and liq20:80 samples. Appendix A show examples of chronograms obtained for the other two PG/VG liquid solutions. Table 1 summarizes the results of experiments of saline and liquid aerosols. In particular, the table reports average values and deviations obtained by several repetitions of measurements.

Regarding the saline aerosol, it is evident that before heating, the average value of frequency shift ($\bar{\Delta f\;}$) does not follow the trend of salt concentration in the solutions. In fact, the $\bar{\Delta f}$ should be higher for the more concentrated saline solution and lower for the one diluted by a factor of ten. Additionally, the average frequency value shows a significant standard deviation, reflecting the possibility of different surface dynamics processes at the interface (e.g., the crystal electrode surface and sample). These dynamics could contribute to lower reproducibility of the deposition when both solid and liquid phases are present.

After heating cycles, the $\bar{\Delta f}$ values not only align with the saline concentration trend in different solutions (NaCl phy > NaCl 1:2 > NaCl 1:10) but also exhibit a lower standard deviation compared to the previous measurements. This indicates improved reproducibility of measurements after heating cycles.

In the case of liquid aerosols, the observed $\bar{\Delta f\;}$ values might be mistakenly attributed to the measurements of solid particles. However, after heating, restoration of the initial frequency indicates the absence of residual mass on the surface of the H-QCM, associated with the evaporation of the liquid. This information obtained by using H-QCM could be used as important feedback for OPC measurements. In simpler terms, if, after heating cycles, the frequency shift persists, the OPC data are correct and are not affected by the presence of a liquid phase of aerosol. In future work, we will analyze the performances of the H-QCM (e.g., sensitivity, the limit of detection, reproducibility, etc.) to assess the possibility of correcting the OPC output data when expressed in terms of mass concentration (µg/m^3^).

The proposed OPC+H-QCM device could be employed in the field of PM measurement, particularly in environments with marine aerosol or fog, which limits the use of simple OPCs without the use of an air sample treatment system before performing the count. Moreover, such systems require minimal space and energy for their operation. 

In the chemical processing industry, exhaust fumes from process reactor stacks simultaneously contain aerosols with different phases. The OPC+H-QCM device could be useful for analyzing these fumes, enabling a thermogravimetric analysis (TGA) alongside total particulate matter dimensional counting.

Although this study presents preliminary results, the proposed tool could be useful for analyzing the fumes produced by e-cigarettes. It could help in identifying solid phases inhaled by users and correlating the quantity and sizes of the particles to some potential pathologies.

## 4. Conclusions

This study investigated the responses of an OPC coupled with an H-QCM to saline and liquid aerosols. Regarding the saline aerosol, the OPC demonstrated that the CMD values for NaClphy and NaCl 1:2 were comparable within experimental error, suggesting similar size distributions, while NaCl 1:10 exhibited a slightly different trend, potentially attributed to dilution effects. The H-QCM, focusing on the saline aerosol, elucidated the dynamics during and after heating. In particular before heating, the frequency shifts suggested the coexistence of solid and liquid phases, influenced by surface phenomena such as deliquescence and aggregation. After heating, the separation of solid and liquid phases became evident, resulting in improved reproducibility in frequency shifts and allowing for a clearer interpretation of the deposition process. In the case of liquid aerosols (PG/VG mixtures), the OPC provided consistent CMD values across different ratios, emphasizing the need for density corrections when OPC is used alone. Conversely, the H-QCM demonstrated its ability to discriminate between solid and liquid phases. Overall, this study highlights the complementary strengths of OPC and H-QCM in aerosol analysis. While the OPC is utilized for size distribution characterization, the H-QCM provides real-time insights into phase discrimination aerosols. These findings contribute to the refinement of aerosol measurement using a low-cost sensor, fostering a more comprehensive understanding.

## Figures and Tables

**Figure 1 sensors-24-02500-f001:**
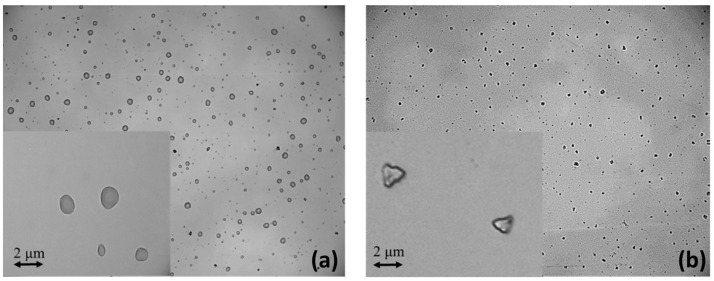
Microscope images of liquid droplets of PG:VG (**a**) and solid particles of NaCl deposited over quartz crystal slice after aerosolization (**b**).

**Figure 2 sensors-24-02500-f002:**
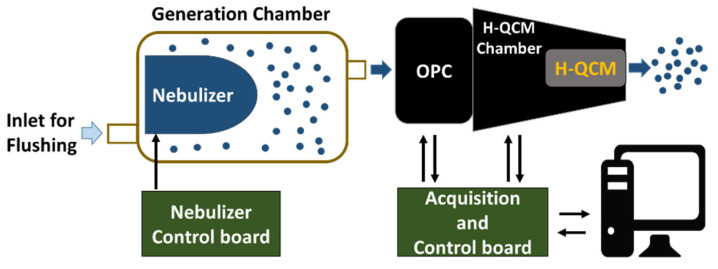
Measurement setup schematic block. The nebulizer was based on ultrasonic vibrating mesh, and the OPC outlet was connected to H-QCM by a suitable adapter (H-QCM chamber).

**Figure 3 sensors-24-02500-f003:**
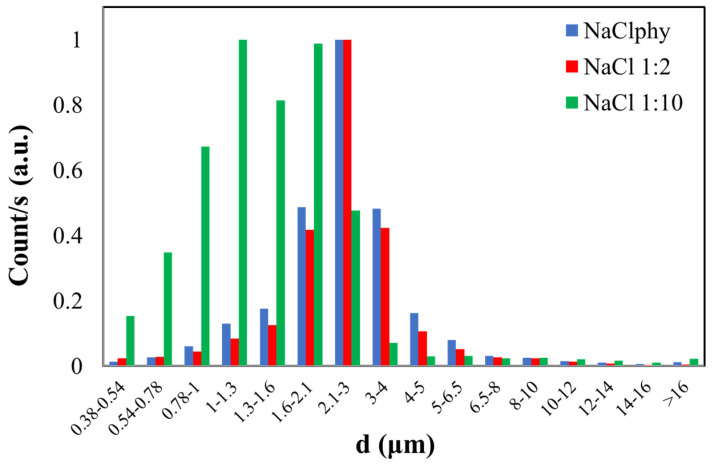
Normalized graph of counts as a function of the aerodynamic diameter (d) of NaCl particles with different concentrations (NaCl_phy_, NaCl_1:2_, and NaCl_1:10_).

**Figure 4 sensors-24-02500-f004:**
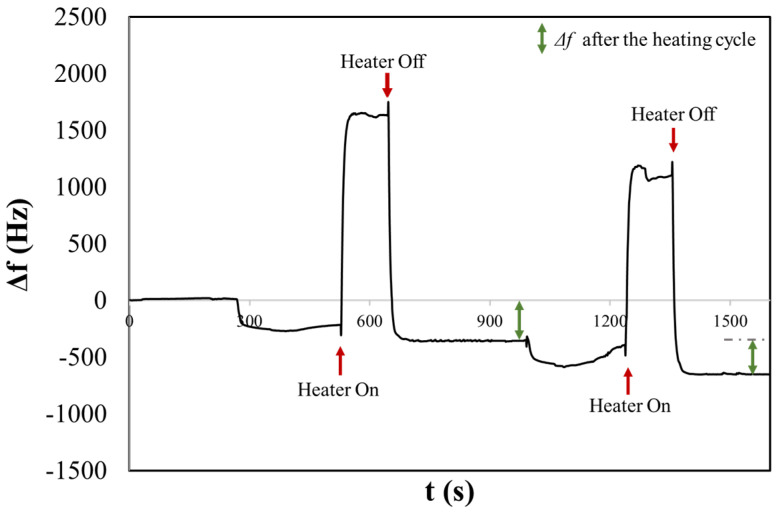
The chronogram of NaCl physiological aerosol shows two consecutive depositions before and after heating on a H-QCM.

**Figure 5 sensors-24-02500-f005:**
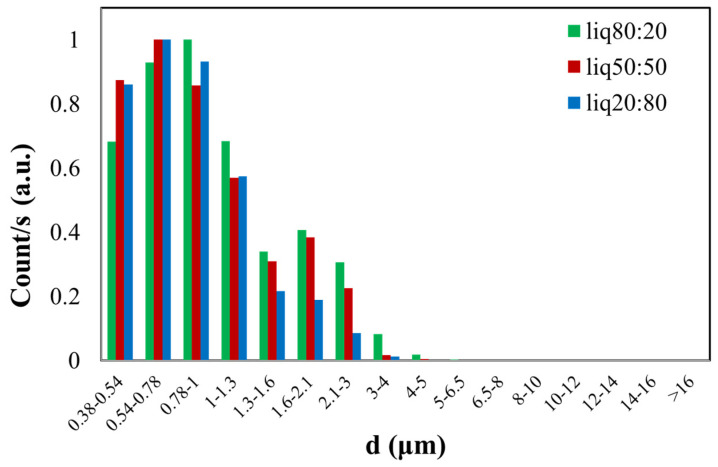
Normalized graph of counts as a function of the aerodynamic diameter (d) of e-liquid droplets with different PG/VG ratios (liq80:20, liq50:50, liq20:80).

**Figure 6 sensors-24-02500-f006:**
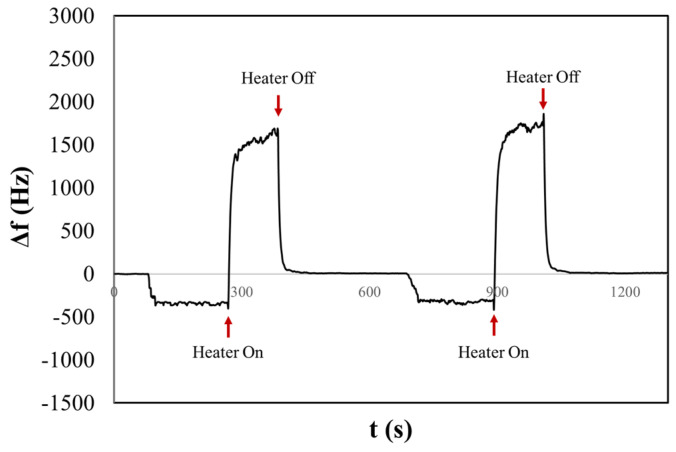
The chronogram of PG/VG 50:50 shows two consecutive depositions before and after heating on a H-QCM.

**Table 1 sensors-24-02500-t001:** Summary of the average frequency shifts for saline and liquid aerosols.

	Before Heating $\bar{\Delta f}$ (Hz)	After Heating $\bar{\Delta f}$ (Hz)	CMD (µm)
**NaCl phy**	219 ± 105	358 ± 40	2.90 ± 0.45
**NaCl 1:2**	112 ± 125	194 ± 21	2.83 ± 0.44
**NaCl 1:10**	381 ± 258	96 ± 30	1.75 ± 0.39
**liq 80:20**	161 ± 5	/	0.90 ± 0.17
**liq 50:50**	337 ± 13	/	1.01 ± 0.15
**liq 20:80**	441 ± 29	/	1.13 ± 0.15

## Data Availability

All data that support the findings of this study are available after the reasonable request to the corresponding author.

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
