# Peer review of "Improving Aerosol Characterization Using an Optical Particle Counter Coupled with a Quartz Crystal Microbalance with an Integrated Microheater"

_sensors, 2024, doi:10.3390/s24082500_

Round 1

Reviewer 1 Report

Comments and Suggestions for Authors

There is no doubt that this paper is very useful for specialists studying the physics and chemistry of the atmosphere and aerosols. The manuscript presents a study on the use of an optical particles counter (OPC) coupled with a quartz crystal microbalance with an integrated microheater (H-QCM) for aerosol measurement. The research is particularly relevant given the significant impact of aerosols on atmospheric pollution, climate, and human health. In general, the text of the article, starting with the abstract, is replete with excessive description of irrelevant details. The paper, which is based on four graphs (figures 5-8) is too detailed and some descriptions resemble a laboratory journal or a thesis text. This remark is certainly of a recommendatory nature. But the reduction of irrelevant points would only increase the positive perception of this paper by other researchers. I think this paper is quite interesting and meets the requirement for publication in Sensors. However, a minor revision is needed before its acceptance.

Some minor remarks:

Figure 3. Photograph of the developed sensor system (a). The H-QCM was fixed by a support in the H-QCM measurement chamber (b) - is of no value to the article and may be deleted or transferred to Supplementary Materials.

The calibration temperature dependence presented in Figure 4 can also be transferred to Supplementary Materials as it does not contain any scientifically relevant information.

Note the arrangement of references, e.g. "The detection of salt particles depends on several factors [42, 43] including the size and concentration of the particles and the refractive index", here the references should either be at the end of the sentence or after each factor you are focusing on.

In Figure S2, values like 9916000 Hz can be converted to 9916 kHz.

This paper would be of much more interest if the method presented by the authors were compared with other existing aerosol measurement methodologies.

Author Response

We are grateful to the reviewer for the valuable questions which will improve and complete our manuscript. We have reported in the following the reviewer questions (Q) and the answers/comments (A). Each text change has been reported in red in the revised manuscript.

Before answering to the reviewer, we have corrected the “x axe” label for the OPC measurements (figure 3 and figure 5 of the revised document version). The error was caused by an artifact of the used graph layout.

Q) Figure 3. Photograph of the developed sensor system (a). The H-QCM was fixed by a support in the H-QCM measurement chamber (b) - is of no value to the article and may be deleted or transferred to Supplementary Materials. 

A) As suggested by the reviewer, we moved the Figure 3 (now Figure S2) in the supplementary materials within its caption rearranging the text.

Q) The calibration temperature dependence presented in Figure 4 can also be transferred to Supplementary Materials as it does not contain any scientifically relevant information.

A) As suggested by the reviewer, we moved the Figure 4 (now Figure S3) in the supplementary materials within its caption rearranging the text.

Q) Note the arrangement of references, e.g. "The detection of salt particles depends on several factors [42, 43] including the size and concentration of the particles and the refractive index", here the references should either be at the end of the sentence or after each factor you are focusing on.

A) As suggested by the reviewer we have corrected the references position.

Q) In Figure S2, values like 9916000 Hz can be converted to 9916 kHz.

A) As suggested by the reviewer we modified the Figure.

Q) This paper would be of much more interest if the method presented by the authors were compared with other existing aerosol measurement methodologies.

A) We are grateful to the reviewer for the valuable comments. Some methods to measure the aerosol (or PM) were  outlined in the introduction. This paper is focused on enhancing the OPC method for aerosol measurement, particularly in scenarios where aerosol may exist in different phases simultaneously, such as liquid and solid. For example, when the OPC is utilized to measure solid particles suspended in air (PM) in the presence of fog (liquid aerosol), the OPC counting  may be influenced by the FOG. As a future outlook, we plan to conduct a comparative study between our device and a simple OPC in a real measurement campaign. The results of this comparison will be published in another paper.

Reviewer 2 Report

Comments and Suggestions for Authors

The manuscript provides a comprehensive investigation into the responses of the OPC and H-QCM to saline and liquid aerosols, offering valuable insights into aerosol analysis techniques. The study is well-structured, with clear subsections detailing the methods, results, and discussions. The integration of experimental data and theoretical explanations enhances the understanding of aerosol dynamics and measurement principles. The use of figures and tables effectively illustrates the experimental setups, results, and comparisons, aiding in the interpretation of findings. The conclusions drawn from the results are well-supported and align with the objectives of the study. Overall, the manuscript presents valuable contributions to the field of aerosol analysis and could be further strengthened with minor revisions and clarifications.

1. Could the authors provide more explanations on how OPC detects particles through light scattering to enhance readers' understanding of its operation?

2. Could the authors further clarify how the duration of 1 minute for aerosol sampling was determined? Was this duration based on previous studies or experimental considerations?

3. Could the authors elaborate further on the broader practical implications of the findings for advancing aerosol measurement technologies and their applications in various fields, such as environmental monitoring, health studies, or industrial processes?

4. Please carefully check the manuscript again to avoid typos such as the superfluous full stop mark in the sentence “…based on light scattering [25, 29.].” (Line 119).

Comments on the Quality of English Language

Minor editing of English language required

Author Response

We are grateful to the reviewer for the valuable questions which will improve and complete our manuscript. We have reported in the following the reviewer questions (Q) and the answers/comments (A). Each text change has been reported in red in the revised manuscript.

Before answering to the reviewer, we have corrected the “x axe” label for the OPC measurements (figure 3 and figure 5 of the revised document version). The error was caused by an artifact of the used graph layout.

Q) Could the authors provide more explanations on how OPC detects particles through light scattering to enhance readers' understanding of its operation?

A) As suggested by the reviewer we have inserted a  working principle in the supplementary materials after the description of Figure S1.

Q) Could the authors further clarify how the duration of 1 minute for aerosol sampling was determined? Was this duration based on previous studies or experimental considerations?

A) The 1 minute is a suitable duration time to have a evaluable signal output from QCM considering different characteristics of the measurements setup as: volume of generation chamber, quantity of generated aerosol, OPC fan performance (in terms of L/min) and OPC saturation level.

Q) Could the authors elaborate further on the broader practical implications of the findings for advancing aerosol measurement technologies and their applications in various fields, such as environmental monitoring, health studies, or industrial processes?

A) We are grateful to the reviewer for his valuable comments aimed at  enhancing the paper content. We have added, in the “results and discussions” section, some possible applications of the presented device.

Q) Please carefully check the manuscript again to avoid typos such as the superfluous full stop mark in the sentence “…based on light scattering [25, 29.].” (Line 119).

A) As suggested by the reviewer we have checked and corrected some text errors, as like eventual typos.

Reviewer 3 Report

Comments and Suggestions for Authors

The manuscript is devoted to the testing of instrument combines the OPC and QCM techniques for aerosol characterization. The Introduction provides sufficient information but there are a few moments need to be improved in other manuscript sections.

Reviewer’s recommendations, questions as well as inaccuracies found are listed below.

lines 99, 105, 237, 248 – text-indent should be added

line 120 – unnecessary line

lines 157, 272, 308 – H2O subscript

Figures in Supplementary do not correspond to description. (line 373, for instance).

Three salt solutions investigated have different aerodynamic diameter distribution according to Figure 5. The way wider so-called “bell” takes place for tenfold diluted solution. Why do CMD values for three samples have the similar confidence interval (±value)? The blank experiment’s data (distilled water) should be added and discussed.

The difference between frequency shift before and after heating was explained with the QCM limitation for relatively big crystals (lines 303-309). The presence of such crystals in aerosol particles gives rise to doubts since the sodium chloride solution is homogenous medium. The authors have mentioned this in text (lines 283-284). This matter should be revealed in details for better understanding. Do the aerosol particles contain some crystals?

Table 1 should be complemented with size parameters obtained.

Author Response

We are grateful to the reviewer for the valuable questions which will improve and complete our manuscript. We have reported in the following the reviewer questions (Q) and the answers/comments (A). Each text change has been reported in red in the revised manuscript.

Before answering to the reviewer, we have corrected the “x axe” label for the OPC measurements (figure 3 and figure 5 of the revised document version). The error was caused by an artifact of the used graph layout.

Q) lines 99, 105, 237, 248 – text-indent should be added, line 120 – unnecessary line, lines 157, 272, 308 – H2O subscript

A) As  suggested by the reviewer, we have checked and corrected the text formatting. The “H2Odist” is a label used as acronym for distillated water. While the “2” character in H2O was in subscripted format but the “Palatino linotype” fonts was used, as suggested by MDPI  guidelines.

Q) Figures in Supplementary do not correspond to description. (line 373, for instance).

A) We have corrected the sentence with “ Figures S8 and S9 in supplementary material show an example of chronograms obtained for “ (line 352, in the reviewed version).

Q) Three salt solutions investigated have different aerodynamic diameter distribution according to Figure 5. The way wider so-called “bell” takes place for tenfold diluted solution. Why do CMD values for three samples have the similar confidence interval (±value)? The blank experiment’s data (distilled water) should be added and discussed.

A) The distributions shown in Figure 3 (of the revised version) are very similar only for NaClphy and NaCl1:2 (red and blue histograms), as they both exhibit a maximum value at 2.1- 3 µm.  Conversely, NaCl 1:10 has a maximum at 1.6 – 2.1 µm (green histogram). Based on the calculated CMDs and their respective error values, it is evident that only NaClphy and NaCl1:2 have a comparable CMD value within the experimental error. In contrast, NaCl1:10 presents a different CMD, even when considering the error, which could be attributable to the high dilution that may cause a different coagulation process and generate small droplets (of solution) along with fine particles. The role of distilled water may be considered negligible, as shown in Figure S5 of the supplementary materials, where is reported an example of the counts (not normalized CMD), as a function of the aerodynamic diameter (d), of NaCl 1:10 and distilled water (H2Odist).

Q) The difference between frequency shift before and after heating was explained with the QCM limitation for relatively big crystals (lines 303-309). The presence of such crystals in aerosol particles gives rise to doubts since the sodium chloride solution is homogenous medium. The authors have mentioned this in text (lines 283-284). This matter should be revealed in details for better understanding. Do the aerosol particles contain some crystals?

A) NaCl solution exists in its ionic form, consisting of Na+ and Cl- ions. Before heating, when NaCl aerosol is deposited onto the H-QCM substrate, it is typically surrounded by water molecules. During this phase, the salt assumes a semi-solubilized state, and the frequency signal does not stabilize  due to the lacks of a distinct crystalline structure (optical microscopic differentiation between PG/VG and NaCl droplets is challenging before heating). The crucial  transformation occurs upon heating, facilitating ion rearrangement and subsequent crystallization. In future research, we could explore these phenomena more thoroughly using this setup, with a specific focus on understanding liquid-crystal phase transitions, deliquescence phenomena and particle aggregation.

Q) Table 1 should be complemented with size parameters obtained.

A) As suggested by the reviewer, we have added a new column (Tab.1) displaying the count mean diameter (CMD) of aerosol particles measured by OPC.